

# Back to pristine levels: a meta-analysis of suspended sediment transport in large German river channels.

Thomas O. Hoffmann[1], Yannik Baulig[1], Stefan Vollmer[1], Jan Blöthe[2], Peter Fiener[3]

[1]Bundesanstalt für Gewässerkunde, 56068 Koblenz, Germany
[2]Department of Geography, University of Freiburg, Schreiberstraße 20, 79098 Freiburg, Germany
[3]Institute of Geography, University of Augsburg, Alter Postweg 118, 86165 Augsburg, Germany

*Correspondence to*: Thomas O. Hoffmann (thomas.hoffmann@bafg.de)

**Abstract.** Suspended sediment is an integral part of riverine transport and functioning that has been strongly altered during the Anthropocene due to the overwhelming human pressure on soils, sediments and the water cycle. Understanding the controls
of changing suspended sediment in rivers is therefore vital for effective management strategies. Here we present results from a trend analysis of suspended sediments covering 62 monitoring stations along the German waterways with more than 440 000 water samples taken between 1990 and 2010. Based on daily monitoring of suspended sediment concentration (SSC), we found significant declines of mean annual SSC and annual suspended sediment loads at 49 of 62 monitoring stations between 1990 and 2010. On average SSC declines by -0.92 mg $l^{-1}yr^{-1}$. At some stations decreases during the 20 years represent up to 50% of
the long-term average SSC. Significant decreases of SSC are associated with declining SSL loads. The contemporary suspended sediment loads of the Rhine at the German-Dutch border approaches the natural base level of ~1 Mt $yr^{-1}$, which was achieved by the Rhine during the mid-Holocene when the suspended sediment load was adjusted to the Holocene climatic conditions and before the onset of increased loads due to human induced land use changes in the Rhine catchment. At this point we can only speculate regarding potential reasons for a decline in sediment supply to larger rivers. We argue that changes
in soil erosion within the catchments and/or the sediment connectivity in upstream headwaters, e.g. due to the construction of small rainwater retention basins, are the major reason for declining SSC in the studied river channels.

## 1 Introduction

Suspended sediment transport from land to ocean is a key component of the global sediment budget that strongly changed in response to human impacts during the Anthropocene. Recent estimates of the pre-Anthropocene magnitude of the land-ocean
transfer range around ~15 Gt per year with suspended sediment transport representing the largest fraction (~14 Gt $yr^{-1}$) (Syvitski et al., 2022; Syvitski and Kettner, 2011). Human-induced land cover changes accelerated hillslope erosion compared to natural background rates by several orders of magnitude (Golosov and Walling, 2019; Montgomery, 2007b; Nearing et al., 2017), strongly increasing the supply of fine sediments to river systems in large parts of the world. During the same time, increased sediment supply is counter-balanced by sediment retention due to the rapidly increasing number of large dams
(Vörösmarty et al., 2003), disrupting the flow path of almost all large rivers in the world, with less than 23% of the global





rivers flowing uninterrupted to the ocean (Grill et al., 2019). Global sediment retention in reservoirs increased by a factor of ~23 from 2.8 Gt yr$^{-1}$ in 1950 to 65 Gt yr$^{-1}$ in 2010 (Syvitski et al., 2022). Owing to the large retention of sediment behind dams, global sediment supply to the oceans ceased by ~50% to about 7.3 Gt yr$^{-1}$ in 2010.

The evolution of regional sediment budgets during the Anthropocene may strongly deviate from the global figures due to
multiple trajectories of socio-environmental changes around the world. While countries with strongly increasing population typically show intensified erosion (Golosov and Walling, 2019) and sediment retention in reservoirs (Annandale et al., 2018), high-income countries start to demount large dams and reestablish the sediment continuity in river systems as a necessary prerequisite of healthy riverine ecosystems. A prominent example for dam removal is the Elwha River restoration project in Washington state (USA), which started in 1992 and resulted in a fully reconnected river after the removal of the last dam in
2014 (East et al., 2018). In Europe, where more than one million barriers fragment rivers (Belletti et al., 2020), efforts are undertaken to reestablish the sediment connectivity and natural functionality of rivers to achieve a good ecological status or potential as requested by the European Water Framework Directive (EC-WFD, 2000). Where possible this includes the removal of dams, esp. in Finland, Sweden, France, Spain and the United Kingdom where almost 5000 old, abandoned or out of use dams were removed (see map on https://damremoval.eu). However, dam removal is hardly possible along waterways.
Here sediment management plans were developed to maintain navigability, hydro power, flood protection and other usages, while reducing the negative effects of dams and to increase the (sediment) connectivity without dam removal (BAW and BfG, 2016).

Additionally, soil conservation programs are implemented in many of regions worldwide with the aim to reduce the loss of soils and to mitigate negative on- and off-site effects of accelerated human-induced soil erosion. The first nation-wide soil
conservation program was started in the USA with the foundation of the Soil Erosion Service in response to the damages of the Dust Bowl in the American prairies in the 1930 ties (Montgomery, 2007a). Following the implementation of large scale soil conservation measures, soil erosion rates strongly decreased from the 1930s to the 1990s (Trimble, 1999).

In China, large-scale vegetation restoration projects started in 1999. In particular, the Grain-for-Green program which established a perennial vegetation cover in the Loess plateau resulted in decreased soil erosion rates in the Yellow river
catchment after 1999 (Wang et al., 2015). However, it is important to note that measures taking highly productive but erosion prone areas out of agricultural use might result in enhanced production and erosion in other regions of the world compensating reduced crop production in China.

In Germany the Federal Soil Protection Act, which was passed in 1999 (BMUV, 2002), builds the basis for the large scale implementation of soil conservation measures. In Germany the implementation of soil conserving agricultural management
slowly increased during the last decades. In 2009/10 about 38% of arable land were under soil conservation (reduced, no inversion tillage), while only about 1% was under no-till (DESTATIS, 2011). However, the tendency of increasing agricultural soil conservation practice is counteracted by a number of processes, which increase the erosion and sediment transport potential. These are (i) a mean increase in rainfall erosivity in Germany between periode 1960 to 1980 and 2001 to 2017 of 66% (Auerswald et al., 2019), (ii) an increase in erosion prone maize cultivation mainly due to biogas production by 70.3%



between the periode 1991 to 2000 and 2012 to 2017 (total area of about 10 500 km²) (DESTATIS, 2021a, b), and (iii) an increase in sediment connectivity due to increasing field sizes and an associated loss in linear landscape feature between fields, which are best approximated via the substantial increase average in farm size from about 39 ha in 1991 to 67 ha in 2016 (BMLE, 2021). Overall, the effect of potential changes on the sediment supply to rivers and their effect on suspended sediment in Germany remains unknown so far.

In this study we use data from the long-term suspended sediment monitoring network that is maintained by the German Waterways and Shipping Authority for sediment management purposes to study the changing suspended sediment dynamics during the last decades. The monitoring started in the 1960s based on work-daily sampling of suspended sediment at ~60 monitoring stations along the German waterways to secure navigability in the context of efficient sediment management. The dataset provides valuable information to study long-term changes of the suspended sediment and to study the control of land

use, river management and climate change on suspended sediment dynamics in Germany. So far, no systematic study of changing suspended sediment concentrations and loads has been undertaken. Here we aim to detect changing suspended sediment concentrations (SSC) between 1990 and 2010 and discuss potential drivers for the observed changes. The time scale from 1990 and 2010 was chosen because most monitoring stations were active during this time period and therefore allows to compare the trends between the different stations.

In this study we mainly focus on SSC instead of suspended sediment loads (SSL), as the former is the primary characteristic of the river system, and the latter is calculated based on the product of SSC and discharge (Q). In most river systems SSC is strongly conditioned by Q but it is an independent variable that does not require the estimation of Q. Thus, we argue that changing SSC is an immediate response of changing sediment sources and dynamics and less likely due to climatic driven changes in Q.

## 2 Methods

### 2.1 Suspended sediment monitoring

SSC in German inland waterways has been monitored using work-daily water samples taken manually by the Federal Waterways and Shipping Administration (Wasserstraßen- und Schifffahrtsverwaltung des Bundes, WSV). SSC monitoring started in 1963 at Hitzacker (Elbe River) and in 1964 in Maxau (River Rhine). Further stations were added to the monitoring

network in the 1960s (10 stations), 70s (23 stations) and in the 80s (17 stations). In East Germany (i.e. in the former Democratic Republic of Germany) monitoring began only in November 1991 adding 20 stations, along the Elbe, Oder, Havel and Spree. By 2020, many monitoring stations provide long-term records that cover more than 30 years. Due to the decommission of some stations after 2000, the maximum number of stations were maintained during 1991 and 2010. For the ease of comparability of the calculated trends between stations, we focus the trend analysis for all stations on this time interval despite

the fact that single stations have much longer monitoring intervals. For selected stations we discuss the stability of the trend analysis in comparison to extended time-periods.





At each monitoring site, 5-liter water samples are taken each work day (i.e. excluding weekends and public holidays). During floods the sampling frequency is increased to up to 3 samples per day, unless sampling was prohibited due to safety reasons. At some stations sampling gaps resulted from shortages of the technical staff or from technical issues. Typically, data gaps
during weekends and holidays (i.e. shorter or equal than two days) were filled using linear regression. Larger data gaps were not considered in the trend analysis of SSC. Water samples are filtered using commercial coffee filters, which are weighed before and after filtering in dry conditions to calculate the daily SSC (mg l$^{-1}$ = g m$^{-3}$). The use of coffee filters is cost-efficient and facilitates measuring SSC at a large number (i.e. ~70 samples per day at the national scale) and of sufficient quality. Calculated SSC values presented in this study include both the mineral and organic material of suspended sediment and are
therefore equivalent to the concentration of the total suspended solids (for more details on sampling of suspended sediment see Hoffmann et al., 2020).

Work-daily SSC time series for each monitoring station covers variable times due to variable start of the monitoring at each site, decommissioning of monitoring stations and larger data gaps due to maintenance issues. Therefore, we only analyzed the long-term SSC-trends of those stations that cover more than 15 years with more than 150 samples per year between 1990 and
2010 resulting in 62 monitoring stations that are included in this study. The monitoring stations are located along 18 waterways in Germany, including the rivers Danube, Rhine, Ems, Weser, Elbe, Oder, and their larger tributaries (Fig.1). The gauging stations cover contributing areas from 2,076 to 159,555 km² (Tab. S1). The topography of the river catchments includes the steep high mountain terrain of the European Alps (e.g. Alpine Rhine and Danube) as well as the mountainous regions with various geological settings in Central Europe and the flat terrain of Northern Germany, which is mainly composed of glacial
and fluvial Quaternary deposits. Daily discharge for each station was taken from the water information system (WISKI) of the Federal Waterways and Shipping Administration, from gauging stations at or nearby the suspended monitoring station. The long-term annual average discharge of the stations ranges from 7.5 to 2261 m³s$^{-1}$ (Tab. S1).

Annual suspended sediment loads (*SSL* in t yr$^{-1}$) were calculated based on the discharge-weighted averaging according to Walling (1981):


$$SSL = k \ \frac{\sum_{i=1}^{n} \overline{Q_i} \times \overline{SSC_i}}{\sum_{i=1}^{n} \overline{Q_i}} \ \bar{Q} \tag{1}$$

where $k = 60^2 \times 24 \times 10^{-6} = 0.0864$ is a unit conversion factor (translating mg l$^{-1}$ to t day$^{-1}$), $\overline{Q_i}$ and $\overline{SSC_i}$ are daily averages of discharge and SSC, respectively, during days with SSC-measurements and $\bar{Q}$ is the annual average discharge. Eq.1 considers
time gaps without SSC measurements and interpolates them based on the ratio of the mean annual discharge ($\bar{Q}$) and the average discharge during days with SSC-observations. As mentioned above, years with less than 150 measurements (e.g. due to operational issues at the monitoring station) were ignored in the trend analysis.

Infrequent sampling or large data gaps may result in strong underestimation of the annual average SSC or the annual suspended load especially in small and flashy river systems (Horowitz et al., 2015; Moatar et al., 2006; Walling and Webb, 1981). The





monitoring stations considered in this study are all located along larger rivers that show a smoothed and buffered behavior compared to smaller river systems (Slabon and Hoffmann, submitted) and no major data gaps were observed during severe floods. Therefore, we are confident that data gaps did not strongly alter the conclusions drawn from our analysis.

**2.2 Trend analysis**

The trend analysis of the SSC from 1990 and 2010 was performed on annual average SSCs for hydrological years starting with the flood period at 1st November and ending at 31st October. We applied linear least-squared regression (LSR), Mann-Kendall test and Sen's slope, which are frequently used in trend analysis of suspended sediment transport (Pohlert, 2018; Walling and Fang, 2003). We applied the LSR due to its simplicity, well knowing that i) residuals of the LSR may not be normally distributed (e.g. due to the existence of extreme SSC-values), or ii) mean annual SSC data may show autocorrelation. Thus,

we checked for normality of the residuals using qq-plots with 95% confidence intervals and for autocorrelation using the Box-Pierce test. 12 stations show non-linear residuals of the LSR, indicating that the assumption of LSR is violated ($p_{lin}$-values in Tab. S2 in italic). Furthermore, 46 of the 62 stations show auto-correlation of the mean annual SSC at $p<0.05$ level, indicating limited applicability of the LSR for trend analysis.

        To evaluate the limitations of the linear regression on time series data we used the non-parametric Mann-Kendall test to detect

monotonic trends of SSC for each station. In case of significant trends at a 5% level (i.e. Mann-Kendall's $p<0.05$), we estimate the magnitude of the trend using the Sen's slope. The Mann-Kendall test and the Sen's slope are calculate using the R package 'trend' (Pohlert, 2018). The rank-based Mann Kendall test is suitable for data with non-normal distributions and does not make any assumptions about the type of trend (i.e. linear or non-linear) as long as values are changing monotonically. The calculation of the magnitude using Sen's slope provides robust estimates in the presence of extreme values.

To check for seasonal differences, we calculated trends based on annual averages for the summer months from June to August and winter seasons from December to February. The winter season covers large parts of the flood period and is controlled by prolonged advective rain falls with higher rainfall amounts, while the summer months are characterized by lower rainfall magnitudes (compared to the winter months) but short and intense convective rain falls with high rainfall erosivity (Fiener et al., 2013).

Under the assumption that SSCs change linearly with time ($SSC = a + bt$, where $t$ is time in years since an arbitrary chosen date, and $a$ and $b$ coefficients) and that the discharge $Q$ remains unchanged, changes of SSC translate to changes of the suspended sediment load ($SSL$):

$$SSL(t) = k \times Q \times SSC(t) \tag{2a}$$


$$\frac{dSSL(t)}{dt} = k \times Q \times \frac{dSSC(t)}{dt} = kbQ \tag{2b}$$





with the conversion factor $k = 0.864 * 365.25$ to obtain annual $SSL$ in units of tonnes per year for units of $SSC$ and $Q$ in mg l$^{-1}$ and m³ s$^{-1}$. (compare Eq. 1). Eq. 2b implies a linear decline of annual $SSL$ for negative $b$, as suggested by the trend analysis

in this study (see results on SSC trend).

### 2.3 Trend of rating parameters

In many studies, sediment rating curves of the form $SSC = \alpha Q^{\beta}$ are used to predict $SSC$ (Doomen et al., 2008), or to understand source/transport characteristics in form of hysteresis analysis (Asselmann, 2000; Hoffmann et al., 2020). In this

study, we do not use the rating approach to predict SSC for infrequent measurements or for SSC data gaps, but use the rating coefficients to identify changes is suspended sediment transport conditions. The rating coefficients $\alpha$ and $\beta$ represent the suspended sediment concentration at unit discharge and the steepness of the increase of SSC with Q, respectively. The steepness is often related to the reactivity of river catchments with increasing topography (Syvitski et al., 2000). Changing conditions of suspended sediment in rivers are translated to changes in the rating coefficients (Warrick, 2015). An increase or

decrease of $\alpha$ relates to parallel upwards or downward shift of the rating curve, and thus changing SSC equally at all discharges. Changes in $\beta$ modifies the frequency-magnitude of suspended sediment transport, with increasing steepness raising the effect of large magnitude events, while decreasing steepness shifts the formative events towards smaller discharges (Warrick, 2015). Here we calculate the rating coefficients for each station and for each year with more than 150 measurements per year between 1990 and 2010. Rating coefficients for a certain year were only used for the trend analysis if the log-linear regression analysis

resulted in significant p-values < 0.05. Hoffmann et al (2020) showed that rating curves of almost all monitoring stations used here are characterized by a bi-linear relation in the log-log diagram, with a scale break at discharges close to the geometric mean discharge $Q_{gm}$ of each station. Therefore, we normalized all $SSC$ and $Q$ data by the geometric mean of each station: $(SSC/SSC_{GM}) = \alpha(Q/Q_{GM})^{\beta}$ and used only data for the high-flow regime $Q/Q_{GM} > 1$, which is mainly controlled by sediment supply from hillslopes during surface generating rain fall events. According to Warwick (2015) the use of normalized

data minimized the effect of the interrelation of $\alpha$ and $\beta$, which arises if rating curves are estimated on non-normalized data.

### 2.4 Driving factors

A multitude of natural and anthropogenic factors potentially control SSC in German waterways, some of which are inherently difficult to quantify. Changes of SSC are either linked to changes of sediment supply to river systems, or sediment retention

along the flow path within the river channels or their neighboring floodplains.

Here, we tried to grasp the influence of drivers that are frequently discussed in context of changing SSC and SSL in large river systems. With respect to changes in the sediment supply from the contributing catchments, the considered drivers include





changes in i) rainfall amount, ii) river discharge (as an integrative parameter for water supply to river systems) and iii) land cover (i.e. coverage by plants protecting the soil from erosion).

Daily rainfall data are taken from the gridded HYRAS dataset (Rauthe et al., 2013) with a spatial resolution of 1 km² derived from 6200 precipitation stations. The dataset covers the river basins in Germany and parts of the neighboring countries that drain into these basins. Daily rainfall data are aggregated to summer and annual sums for the years between 1990 and 2010 for nine selected river catchments including the Rhine upstream of Maxau, and some of its tributaries (Lahn, Neckar, Main, Moselle), the German part of the Danube catchment upstream of Straubing, as well as the Ems, Weser and Elbe catchments

(see Fig. S1 for location of gauging stations and contributing catchments). Due to the low temporal resolution of the rainfall data, we did not calculate the rainfall erosivity, which is strongly conditioned by short and intense rainfall events, as daily rainfall data strongly smooth maximum rainfall intensity. Instead, we used the rainfall erosivity maps for Germany representing the central years of 1975 (Sauerborn, 1994) and 2009 (Auerswald et al., 2019) to derive the trend of rainfall erosivity in Germany.

Discharge for all SSC monitoring stations is taken from the Water Information System (WISKI) of the German Waterways and Shipping Authority. Here we used daily discharge data from gauging stations located at or close to the suspended monitoring stations. For the trend analysis, daily discharge data are aggregated to mean annual discharges aggregated using the hydrological year (similar to SSC and SSL).

The trend analysis of the mean annual and summer precipitation per catchment and mean annual discharge for gauging stations
is performed in the same way as for SSC using the Mann-Kendall test, and the magnitude of the Sen's slope.

To consider land use and land management change we used CORINE land cover data provided by Copernicus, the European Union's Earth observation programme (https://land.copernicus.eu/pan-european/corine-land-cover). Here we used the rasterized land cover data (Version 2020_20u1) with a spatial resolution of 100 m x 100 m for years 1990, 2000, 2006, 2012 and 2018, to derive i) artificial areas, ii) arable land, iii) forest and iv) pasture. The data of each year were clipped with the

polygons derived for the nine river catchments (Fig. S1) and the coverage for each of the 44 land use classes estimated for the nine river catchments. Artificial areas were derived from the sum of CORINE-classes 1 to 11, arable land from the sum of CORINE-classes 12 to 17, and forest from the sum of CORINE-classes 23 to 25. Pasture is given by CORINE-class 18. These land use classes cover the majority of the considered catchments with remaining land uses range between 1,4 to 19 % for the various years and river catchments (excluding the Rhine catchment upstream of Maxau). The Rhine catchment shows a

somehow larger fraction of other land cover classes (approximately 25%) due to high fraction of alpine areas in this catchment. Drivers affecting sediment retention in river systems are represented here by the volume of reservoirs in Germany (taken from Deutsches Talsperrenkomitee e.V., 2013) and the numbers of dams along the German waterways. The cumulative increase of both numbers between 1900 and 2010 is used as a first order proxy for the timing of sediment retention. While reservoirs are typically located in smaller headwater catchments, their cumulative volume is a first order proxy of disconnectivity of sediment

supply from hillslopes to river channels. In contrast, dams along the waterways (constructed mainly to produce hydro power



and to improve navigation) retain suspended sediments within the river channel. Their trapping efficiency in terms of suspended sediments is typically much lower than that of reservoirs.

## 3 Results

### 3.1 SSC trends

For both trend-algorithms (i.e. linear regression and Sen's slope) 56 stations showed significant changes of mean annual SSCs between 1990 and 2010 (Fig. 2+3 and Tab. S2). Only 6 stations show no significant change at the 5% level. SSC-gradients of stations with significant changes derived from the linear regression ($b_{lin}$) range between -2.02 to -0.38 mg l$^{-1}$yr$^{-1}$ with a mean of -0.92 mg l$^{-1}$yr$^{-1}$. Relative to the mean SSC the gradients represent declines of -0.2 to -8.7 % yr$^{-1}$. Sen's slopes ($b_{sen}$) for station with significant p-values at 5% level range in the same order between -1.99 to -0.37 mg l$^{-1}$yr$^{-1}$ (or -8.9 to -0.26 % yr$^{-1}$)

with a mean of -0.92 mg l$^{-1}$yr$^{-1}$ (3.7 % yr$^{-1}$) (compare Fig. 2).

Negative $b$ values indicate that 56 out off 62 stations are characterized by a significant decline of mean annual SSCs between 1990 and 2010. Both trend algorithms show comparable trends with only marginal differences between the Sen's-slope and the linear gradient (compare also Fig. S02). The similarity of both approaches suggests that in our case autocorrelation of annual averaged SSCs and outliers (due to extremes) do not play a major role for the trend estimation. However, for statistical

correctness we use the Sens's slope for further analysis as residuals of the LSR are mostly not normally distributed, which violates the assumptions of the LSR.

Stations without significant changes for both trend algorithms over the time period 1990 to 2010 include: Straubing (ID 102) and Jochenstein (ID 107) along the Danube River, Weil (202) at the Rhine river (202), Rheine (301) at the Weser river and Hitzacker (502) the most downstream station along the Elbe river (Fig. 3b). Additionally, for linear regression station

Brodenbach (258) at the Moselle River and for the Sens-slope regression station Viereth (231) at the Main River show no significant decreases (see Fig. 3a).

The magnitude of the decline seems to be unaffected by the average discharge at the stations but is weakly controlled by the average SSC (Fig. 4). Stations with high average SSCs show generally stronger declines than stations with low average SSCs. This is especially evident along the Rhine, where SSCs at the stations in the Upper Rhine show lower average SSCs and either

no (stations Weil) or lower (station IDs 203, 205, 206 and 207) declining trends than the stations at the Middle and Lower Rhine (station IDs 212, 215, 216 and 217), where average SSCs are higher and trend magnitudes are larger (Fig. 5). Strongest declines along the German waterways with $b_{sens}$ < -1.5 mgl$^{-1}$yr$^{-1}$ are overserved at four stations in the Elbe and Weser catchments (station IDs 531, 516, 520 and 421). However, Fig. 3 does not indicate a general spatial pattern of the SSC decline. The seasonal trend analysis confirms the general picture of the annual trends (see Tab. S2 and Fig. 5 for examples at Maxau

and Emmerich). 52 of 62 stations show a significant trend of the average SSC during June, July and August (summer trend) from 1990 to 2010. Except for Hitzacker, the most downstream station at the river Elbe, all significant summer trends of the monitoring stations in Germany are negative ranging from -2.87 to -0.36 mg l$^{-1}$yr$^{-1}$ (or -10.6 to -0.3 % yr$^{-1}$). At Hitzacker the



summer trend is increasing with 0.76 mg l$^{-1}$yr$^{-1}$ (or 2.3 % yr$^{-1}$). Interestingly, the largest summer decline is also observed at the river Elbe in Magdeburg ($b_{summer}$ = -2.87 mg l$^{-1}$yr$^{-1}$ or 10.2 % yr$^{-1}$) approximately 190 km upstream of Hitzacker. Average SSCs during the winter months (December to February) decline significantly at 49 of 62 stations, with 13 stations showing no significant change and no station showing a significant increase. $b_{winter}$ ranges -3.55 and -0.39 mg l$^{-1}$yr$^{-1}$ (or -11.6 to -0.2 % yr$^{-1}$), while most station scatter between -1.5 and 0.5 mg l$^{-1}$yr$^{-1}$. At Hitzacker, the winter months show a significant decline with -0.61 mg l$^{-1}$yr$^{-1}$ (or -1.8 % yr$^{-1}$), despite the increase of SSC during the summer months. This contrasting seasonal trends result in insignificant changes during the whole year.

The comparison of the seasonal trends (Fig. 6) indicates that declines during the summer months are somehow larger (i.e. more negative) for 30 stations, and only 10 stations show stronger decreases during the winter. This agrees with the slightly stronger average decrease in the summer months (-1.11 mg l$^{-1}$yr$^{-1}$, or -4.7 % yr$^{-1}$) than in the winter months (on average -0.94 mg l$^{-1}$a$^{-1}$ or -3.7 % yr$^{-1}$).

## 3.2 Trends of rating parameters

We were able to calculate a significant annual sediment rating with more than 15 years per station for 59 stations. 41 stations showed a significant decline of the $\alpha$ coefficient with $d\alpha/dt$ ranging between -1.08 and -0.28 mgl$^{-1}$yr$^{-1}$ (or -6.0 % yr$^{-1}$ and -1 % yr$^{-1}$). The mean and median value of $d\alpha/dt$ for all stations is -0.71 and -0.72 mgl$^{-1}$yr$^{-1}$ (-3.2 % yr$^{-1}$), respectively, which is a little smaller than the average Sen's slope. Strongest declines are observed at the stations along the Rhine (Tab. S2) and none of the 59 stations showed a significant increase of $\alpha$. Only 14 of 59 stations show a significant change of the rating exponent $\beta$; 9 stations show an increase in $\beta$ (increasing reactivity) and 5 stations show a decrease in $\beta$ (decreasing reactivity). Overall $d\beta/dt$ ranges between -0.072 yr$^{-1}$ and +0.078 yr$^{-1}$ (Tab. S2). At some stations with a strongly increasing rating exponent the $\alpha$ coefficient strongly declines, indicating an inverse relationship between changes in $\alpha$ and $\beta$ despite the normalization of the rating analysis.

## 3.3 SSL trends

Only one station (i.e. Marktbreit at the river Main) of 63 stations shows a significant change of the mean annual discharge at 5% significance level based on the Mann-Kendall test (not shown). In conjunction with the declining SSC at the studied monitoring stations, SSL is expected to strongly decrease between 1990 and 2010. 48 of 63 stations show a significant negative trend of SSL with annual declines ranging between 0.1 and 94.9 kt yr$^{-2}$, with an average of 14.7 kt yr$^{-2}$. As suggested by Eq. 2 the decline of SSL grows with increasing discharge (Fig. 7). For instance, the annual load at Maxau declined on average by 28.5 kt yr$^{-2}$ from around 1.3 Mt yr$^{-1}$ to 0.8 Mt yr$^{-1}$ between 1990 and 2010, the decline in Emmerich was twice as high (i.e. 66.3 kt yr$^{-2}$) from 2.8 Mt yr$^{-1}$ to 1.8 Mt yr$^{-1}$ for the same time.



A mean decline of SSC between 1990 and 2010 of $dSSC/dt = b = -0.92$ mg l$^{-1}$yr$^{-1}$ can be used to calculate an annual decline of ~30 kt yr$^{-2}$ per 1 m³s$^{-1}$ of discharge, which is in general accordance with regression of $-dSSL/dt$ and $Q_m$ resulting in a

regression slope of 31.5 kt yr$^{-1}$ per 1 m³s$^{-1}$ as shown by the blue dotted line in Fig. 7.

## 4 Discussion

The presented trend analysis is based on 62 monitoring stations, with an average monitoring interval of 19.9 years and almost 440 000 SSC samples in total. Overall the monitoring stations are characterized by a low fraction of data gaps, with an average rate of only 4% of missing data (excluding the weekends and federal holidays). Due to the large size of the studied rivers, their

variability is reduced compared to much smaller tributary streams. In large rivers, which are considered in this study, floods with increased SSCs and loads last several days, suggesting that daily sampling is sufficient to cover the importance of floods (Slabon and Hoffmann, submitted). In contrast to the calculation of annual loads, which require a continuous sampling, irregularities in sampling frequency less strongly affect the average annual SSC, which is the focus of this study. Thus, we are confident that the data gaps are not substantially affecting our trend analysis. Furthermore, we can exclude that changes in

monitoring techniques are affecting our results, since the measuring and monitoring approach has not been changed since its introduction in the 1960s. Furthermore, for the analysis of the time span between 1990 and 2010 we only used sampling stations without change in sampling location.

As mentioned above, sampling is limited on surface water samples suggesting an underestimation of the cross-sectional average SSC. The point sampling certainly effects the estimation of the total fluxes, but should not affect the trend analysis,

as SSC likely change in the entire cross section and not preferentially at the water surface.

In summary, we are confident that the estimated trends are real and not biased due to the techniques and approaches of the suspended sediment monitoring in the German waterways. Furthermore, the similarity of results derived from the Man-Kendall test in combination with the Sens's slope test and the trends derived from the least square linear regression suggest that calculated trends are robust.


### 4.1 Declining trends of SSC and SSL in German river system

Despite the strong year to year variability of the annual average SSC at each station, our trend analysis shows a widespread declining trend of SSC at 58 out of 62 monitoring stations. No station shows increasing trends of annual average SSCs between 1990 and 2010. This consistent trend is surprising as it covers a broad range of river catchments with different sizes (ranging

between 2000 and 160 000 km²), and variable topographic and geological conditions. Topographic conditions range from mountain topography of the European Alps, which cover parts of the Rhine and Danube catchments, to the lowland topography of Northern Germany, mainly covering the Ems, and northern parts of the Weser and Elbe catchments. Climate conditions range from maritime in West Germany to more continental climates in the East. Furthermore, land use history in West and



East Germany followed different trajectories before and shortly after the reunification in 1990. Despite these differences in
controlling factors, SSC declines consistently without any larger spatial pattern of the observed trends.

Based on the Sen's slope, on average SSC declines by -0.92 mg l$^{-1}$yr$^{-1}$, representing -3.7 % yr$^{-1}$ relative to the long-term
average SSC of each station. For instance, SSC at the station Emmerich located at the Lower Rhine (Fig. 4) declined by
0.95 mgl$^{-1}$each year or in total by approximately 50% from ~40 mgl$^{-1}$ in 1990 to roughly 20 mg l$^{-1}$ in 2010. Declines during
the summer month at this station were somehow larger (-1.21 mg l$^{-1}$yr$^{-1}$) than during the winter months (-0.99 mg l$^{-1}$yr$^{-1}$) (Fig.
4). However, due to the strong interannual variability of the annual mean SSC, the differences between summer and winter
months are not statistically significant.

The station at Hitzacker belongs to the few monitoring stations that do not show a significant decline at the annual scale. Here
declining SSC in the winter months and increasing SSC in the summer months counterbalance each other at the annual scale
(Tab. S2). However, a trend analysis by Hillebrand et al (2018) using the same data, but a longer timescale from 1964 to 2014,
indicated a significant declining trend. This monitoring station is located at the downstream end of the Elbe river, and shows
strong plankton growth during the summer months. Since SSC represents here the total suspended matter (including mineral
and organic particles), increasing SSC during the summer months are mainly controlled by the autotrophic production within
the river channel, and decreasing winter SSC likely results from changing supply conditions, either from reduced soil erosion
and/or reduced sediment connectivity in river channels. While many monitoring stations show evidences of increased
autotrophic production during the summer months (Hoffmann et al., 2020) this effect is most pronounced at the lower Elbe
river. The higher frequency of low-flow conditions during dry and warm summer months in combination with reduced flow
velocities in the impounded sections of the tributary waterways increased the potential for strong algal blooms in in these rivers
as an effect of future climate changes and likely effect future SSC trends.

A positive sediment rating of SSC and discharge at almost all monitoring stations (Hoffmann et al., 2020) suggest that declining
SSC levels might be related to decreasing discharges. The trend of the rating parameters indicate that changes are mainly
related to the rating coefficient $a$, which shows a significant decrease at 41 out of 59 stations. Changes of the rating coefficient
$a$ are related to parallel shifts of the rating curve (Warrick, 2015) and therefore represent the changing SSC at all discharge
levels. Significant changes of the rating exponent $b$, which represents SSC-shifts between high and low magnitude discharges,
are much less pronounced (only 14 of 59 stations show significant changes) and more or less equally distributed between
increased and decreased reactivity. Therefore, we argue that changes in SSC are mainly driven by decreased sediment supply
to river channels or decreased connectivity (e.g. sedimentation within the river channels) within the river network and are not
related to changing discharge conditions. Furthermore, we argue that the strong decline of the suspended load, which is
dominated by fine silts and clays, has no major impact on the channel morphology, which is dominantly formed by sand and
gravels. Therefore, decreases suspended loads had no major impact on sediment management in the German waterways.

In contrast to the general and widespread decreasing trend in the German waterways, trends of suspended sediment transport
in neighboring river systems outside of Germany are more variable. Poulier et al. (2019) studied the annual, flow-averaged
suspended matter concentration (fw-SSC) at six stations in the Rhone catchment in France between 2000 and 2016. The Rhone





catchment is characterized by alpine topography in the North and steep torrential characteristics with Mediterranean climate in the South. Since the end of the Second World war, river management strongly altered the flow characteristics and hydro-

morphology of the rivers. However, none of the stations show a significant change of the SSC between 2000 and 2016 due to the strong interannual variability and the rather short time period. Mean annual SSC in the Warta River (Poland) increased between 1961 and 1980 by 2.6 mg l$^{-1}$ per year (Skolasińska and Nowak, 2018). The increase is mainly attributed to the river channel management (i.e. deepening and cleaning of channel bed), the opening of a large lignite mine and the increased growth of phytoplankton during the summer month in response to higher nutrient levels and rising summer temperatures (similar to

our results from the lower Elbe river).

The stations along the upper Danube River, upstream of the German-Austrian border, are characterized by declining trends (station 105 Vilshofen and 106 Kachlet) or insignificant changes (station 102 Straubing and 107 Jochenstein). Habersack et al. (2016) present marginal declines of the suspended sediment load along the Middle Danube for the time period between 1985 and 2000 compared to loads before 1960 AD. In contrast to small decreases of the suspended load upstream of the Iron

Gate I and II hydropower complex, which were constructed in 1972 and 1984 in Romania, downstream of these reservoirs suspended sediment loads of the Lower Danube massively decreased, with suspended sediment supply to the Black see decreasing from more than 1 Mt yr$^{-1}$ around 1960 AD to 0.4 Mt yr$^{-1}$ from 1985-2000 (Habersack et al., 2016). Therefore, in the lower Danube, strong declines of SSL are related to the primary control of the two dams in the main river channel.

The trend of decreasing SSC in Germany is paralleled by strong declining nutrient and contaminant levels in the river channels

in Germany (REF: UBA 2014). Strong reductions in point and diffusive nutrient inputs – for instance the TOC declined from 5-6 mg l$^{-1}$ in 1990 to ~3 mg l$^{-1}$ in 2010 in the river Rhine at Koblenz (FGG Elbe, 2013) – suggest that declining SSC-levels in the German waterways is related to decreased nutrient supplies. Certainly, the decrease of the nutrients and contaminants do not directly account for decreases of the SSC, since nutrient and contaminant levels are at least an order of magnitude smaller than that of the SSC. However, measures that caused decreased nutrient supply, such as construction of buffer strips between

arable land and river systems or improved waste water treatment, might also affect sediment supplies to river systems (see also discussion on drivers of SSC decrease).

Due to the consideration of total suspended matter, which includes mineral and organic fractions, differences between changes in summer and winter SSC can be partially explained by the effect of decreasing TOC. TOC in the German waterways originates from aquatic biomass-derived organic matter and mineral-associated organic matter originating from eroded

hillslopes (Hoffmann et al., 2020). Biomass-derived organic matter dominates during summer months, which are characterized by low flow conditions with long residence times of biomass in the river system, high availability of light and warm temperatures promoting algal growth and causing higher TOC contents during the summer. Thus, declining TOC between 1990 and 2010 is mainly observed during the summer months, as in the case of the station at Hitzacker, which is strongly influenced by biomass-derived organic matter (Hardenbicker et al., 2014; Hillebrand et al., 2018).






## 4.2 Long-term context of contemporary SSC and SSL changes

In this paper we focus on the timeframe from 1990 to 2010, as most monitoring stations were active during that time and calculated trends are therefore directly comparable to each other. However, looking at longer-term changes is helpful to evaluate the trends and to derive primary controls of these trends.

SSCs changes between 1970 and 2020 are unraveled by the compilation of the residual SSCs, which were calculated using daily SSCs and the long-term average SSC for each station (compare Tab. S1). The average SSC residual of all stations shows a strong variability before 1995 without any major trend (Fig. 8). After 1995, SSC declined for almost all stations with a decreased interannual variability. After 2010, the decline stopped at ~ -7 mg l$^{-1}$ relative to the long-term average SSC of each station. The compilation of the residual SSCs suggest that the declining trend between 1990 and 2010 is mainly cause by a
gradual decline of SSC, which was accentuated between 1995 and 2010 at most stations. Due to the strong variability of the annual SSC residuals, single stations might have a different trajectory than the average behavior (blue line in Fig. 8). However, the consistent behavior of the majority of stations indicate a major change between 1995 and 2010 of the environmental factors controlling the suspended sediment dynamics in the Germany waterways.

To evaluate the long-term context of the contemporary decline of the SSC and SSL beyond the start of suspended sediment
monitoring in Germany, we use reconstructed suspended sediment supply rates to the Rhine delta below the Dutch/German border (Erkens, 2009). The suspended sediment load to the Rhine delta for the last 9000 years stratified into 500 years' time slices was reconstructed by Erkens (2009) using a sediment budget approach based on a detailed stratigraphical analysis of alluvial deposits of the Rhine/Meuse delta. Based on various assumption of the trapping efficiency of the Rhine/Meuse delta, the author relates the deposition of silt and clay to the sediment supply from the upstream contributing catchment. Increased
early Holocene suspended sediment loads ~ 2.3 Mt yr$^{-1}$ are associated to the delayed response of the river Rhine to environmental changes from Postglacial to Holocene conditions (Fig. 9). This transition is completed around 6000 BP, when suspended sediment loads achieve a minimum of around 1 Mt yr$^{-1}$ under a well-established forest cover in the Rhine catchment during the mid-Holocene. Suspended sediment loads at the delta apex coincide with a low geomorphic activity during that time as indicated by a low frequency of dated alluvial deposits in Central Europe (Hoffmann et al., 2009). The increase of the
reconstructed suspended load in the Late Holocene is related to the intensified deforestation and agricultural land use starting during the Bronze Age and continuing during the Roman and Medieval periods. Again, this increase is evidenced by enhanced overbank deposition in Central European floodplains starting in the Bronze age and culminating the de Medieval period (Hoffmann et al., 2009). In summary, low mid-Holocene sediment yields of around 1 Mt yr$^{-1}$ can be considered as the natural baseline during the Holocene before the start of human impact (Fig. 9).

SSLs at the station in Emmerich in the 1980s and early 1990s are in the same order but slightly larger than reconstructed loads during the late Holocene. Due to the temporal gap between the reconstruction and the start of the monitoring it remains unclear when the maximum SSL in the Rhine river was approached in response to the increased human-induced impact. Furthermore, the magnitude of this maximum remains unknown. However, measured annual loads of the Rhine at Emmerich (blue dots in





Fig. 7) indicate that SSL declines from 3-4 Mt yr$^{-1}$ in the 1980s to 1 Mt yr$^{-1}$ in 2010 and therefore approaches the natural
baseline, which represents mid-Holocene conditions without a strong human impact. The similarity of suspended sediment
loads during the mid-Holocene and the contemporary load at Emmerich can be pure coincidence and does not imply that the
river Rhine and its catchment is in a natural or good ecological state. However, it implies that the suspended sediment is
approaching the natural baseline. Furthermore, it raises the question of the driving forces causing the strong declining trend in
the large German river systems.

**4.3 Causes of decreasing SSC and SSL**

Soil erosion is strongly conditioned by the total rainfall amount, or more precisely by its kinetic energy and the intensity of the
rainfall (McGehee et al., 2021). The rainfall amount, which was derived for the nine selected catchments (Fig. S1) with a
cumulative catchment area of 318 000 km² based on daily precipitation HYRAS-data, shows no significant trends for the
summer and annual precipitation sums (Fig. S2). While total rainfall sums may not change, rainfall extremes are intensifying
as Earth's climate warms (Fowler et al., 2021), suggesting that soil erosion will increase in response to climatic changes, as
the number of erosive rainfalls (in Germany defined as rainfalls with more than 10 mm and a maximum 30-min intensity of >
10 mm h$^{-1}$) increases. Long-term records of rainfall erosivity at 10 stations in West Germany indicate an increase in rainfall
erosivity between April and November by 2.1 % per year or 42 % from 1990 to 2010 (Fiener et al., 2013). This growth is in
the same order as the 1.7 % increase per year estimated by Auerswald et al. (2019), who compared the rainfall erosivity map
by Sauerborn (1994), representing the climate conditions in Germany between 1960 and 1980, with their own map derived
from contiguous radar rain data for the central year 2009 (Auerswald et al., 2019). In addition to the general growth of the
annual rainfall erosivity, increases of winter erosivity between October and March affect the cropping factor (C-factor) of the
USLE, as winter rains fall in arable land with a low vegetation cover, furthermore enhancing the erosivity (Auerswald et al.,
2019)

The intensification of rainfall erosivity is in stark contrast to the declining SSC at most monitoring stations in Germany. While
higher rainfall erosivity likely increases the sediment supply to river systems, decreasing SSC in rivers suggest lower sediment
supply to the headwater basins. The decoupling of the SSC trends in the large river systems and increased erosivity might be
explained by the very localized impact of intense convective summer extremes, that result in massive fluxes in small to medium
scale river systems, but hardly affect rivers with contributing catchment areas that are more than 100 times larger than large
convective summer cells. Therefore, we argue that changes in precipitation magnitude and frequency unlikely explain the
declining trends of suspended sediment in the river systems.

Insignificant changes of the annual precipitation in each catchment is in line with the results of the Mann-Kendall test for mean
annual discharges. Only one station (i.e. Marktbreit at the river Main) of 49 stations show a significant change at 5%
significance level. Even when increasing the significance level to 10% only 6 stations (232, 233, 237, 242, 502 and 543) show
significant changes. Close linkages between catchment-average annual precipitation and mean annual discharge (Fig. 10)
indicate that year to year variability of discharge is linked to the inter-annual variability of annual precipitation, which did not





significantly decrease for the considered stations and their contributing catchments between 1990 and 2010 (Fig. 10). Thus, meteo-hydrologically driven changes of SSC and SSL can be excluded as a major driver of suspended sediment dynamics in the German river systems between 1990 and 2010, which is also indicated by the strong differences of SSL between the decade
1990-1999 and 2000-2010 for some of the stations in Fig. 10. This statement is further corroborated by the weak tendency of increasing discharges at some of the considered gauging stations (all of the six stations with significant changes of mean annual discharge at 10% level show a slight increase of discharge), which would rather explain rising instead of declining SSC levels. However, given the strong increases of rainfall extremes during the last decades, SSC declines might have been slowed after 2010 or may even inverted in the near future in response to projected changes (Fowler et al., 2021).

Scientific evidence shows that soil erosion under arable land is orders of magnitude larger than under forest or otherwise natural land cover (Montgomery, 2007b). In Germany the estimated mean erosion potential under arable is 5.2 t ha$^{-1}$yr$^{-1}$, while it is only 0.5 and 0.2 t ha$^{-1}$yr$^{-1}$ under grassland and natural forest, respectively (Auerswald et al., 2009). Generally, it is assumed the erosion rates under forest are even smaller, which is partly counterbalanced as forests are often located on steeper slopes, not suitable for agricultural use. These land use specific erosion rates suggest that the extent of arable land is an important
control for the erosion and sediment supply to river systems. Here we use changes of the extent of arable land and forest cover – the latter is the native vegetation in Central Europe – as a first order proxy for land use driven changes (Tab. 1). While the extent of arable land decreased by several percent in the Lahn, Weser and Elbe catchments (3.3, 3.7 and 7.2%) between 1990 and 2012, other catchments listed in Tab. 1 show only marginal changes, or even increases of the arable land. For instance, in the Upper Danube and Neckar catchments the extent of arable land increased by 4.1 and 3.9%, respectively, implying increases
of catchment-average soil erosion rates. This contradicts the observed SSC trends and cannot be explained by marginal increased forest covers (e.g. in the Danube catchment by 0.4% or in the Elbe catchment by 1.8%) in the same time. Despite the importance of arable land, it remains questionable whether these modest and inconsistent changes of the extent of various land cover classes may explain the overall large-scale decrease of SSC at the majority of the gauging stations.

Another factor potentially leading to a decline in sediment supply is the increase in soil conservation applied on arable land.
Following DESTATIS (2011) 38% of arable land was under soil conservation in 2009/10. Unfortunately, it is not clear to which extent the soil conservation increased between 1990 and 2010 as there is no statistical information available for soil management in Germany before 2009. However, based on the agricultural development after the reunification in 1990, it can be assumed that the proportion of soil conservation practice most substantially increased between 1990 and 2010 in the former eastern part, which in 2009/10 had the largest proportion of soil conservation (53.4 ± 9.52%) (DESTATIS, 2011). Partly the
effect of increasing soil conservation will have been counteracted via the substantial increase in maize production since the early 2000s following the support of biogas production in Germany (Clearingstelle EEG/KWKG, 2002). Overall, the proportion of maize cultivation increased by 70.3% between the periode 1991 to 2000 and 2012 to 2017 (total area of about 10 500 km²) (DESTATIS, 2021a, b).

Another reason for different trends between the former Western and Eastern part of Germany might result from different
developments regarding field layouts and linear landscape structures, e.g. hedges, grass strips. While Eastern Germany was





already in 1990 dominated by large fields with little landscape structures in-between, field sizes substantially increased in many areas of Western Germany, which can be approximated by a mean increase in farm size mostly in Western Germany from 39 ha in 1991 to 67 ha in 2016 (BMLE, 2021). Larger fields generally increase sediment connectivity in arable landscapes (Fiener et al., 2011), and thus, should increase sediment supply to rivers. However, neither the potentially stronger increase in

soil conservation practice in the former eastern part of Germany (which should reduce sediment supply), nor the increase in field sizes and the loss of landscape structures between fields in the former western part of Germany (which should increase sediment supply), result in substantially different trends in SSC between the rivers draining West and East Germany (Fig. 4). Reservoirs strongly decrease the longitudinal connectivity of suspended matter in river channels and are the primary cause of the strong global decline of sediment supply to the world's oceans (Syvitski et al., 2022). The Upper Rhine between Basel

(CH) and Iffezheim (GER) is controlled by 10 barrages. Furthermore, all tributary waterways are managed by sequences of barrages that halt bed load transport and retain substantial amounts of suspended sediment. The barrages along the Upper Rhine retain suspended matter in the order of 0.3 Mt yr$^{-1}$ between the gauging station at Weil and Plittersdorf (Frings et al., 2019; Hillebrand and Frings, 2017), representing 20% of the 1.45 Mt yr$^{-1}$ upstream inflow at Weil. Therefore, it is frequently argued (van der Perk et al., 2019) that suspended sediment load in the Lower Rhine is reduced in response to the construction

of barrages and reservoirs in the catchment. However, barrages along the German waterways were mainly constructed between 1930 and 1940 and between 1955 and 1970 (Fig. 11). Furthermore, the construction of large reservoirs for hydro power generation along the main waterways and for storage of drinking water in the upstream headwaters was almost completed by 1990 and only a small number of upstream reservoirs were built after 1990 in single regions. In many cases, river systems react with immediate, step-wise declining sediment transport rates in response to dam closures (Habersack et al., 2016; Kondolf

et al., 2018; Sun et al., 2016; Walling and Fang, 2003), suggesting that post-damming loads remain constant and low after dam construction.

SSC and SSL changes indicate that there are no abrupt declines of the suspended sediments in the German river systems but continuous declines over at least 20 years, indicating that the supply to the river systems must decrease or sediment retention must increase more or less continuously each year. The decline of the annual SSLs provides a benchmark for the evaluation

of the effects of dams in river systems. For instance, along the Rhine river upstream of Emmerich, SSL declines by 66.3 kt each year between 1990 and 2010. Continued declines in response to dams however requires that each year after dam closure, the amount of sediment deposited in the reservoir must increase. However, as reservoirs fill up with sediment, trapping efficiencies decrease and the amount of deposited in the reservoir likely decrease (Kondolf et al., 2014, Brune, 1953 #940, Brune, 1953 #940). Therefore, we argue that the construction of barrages and reservoirs unlikely are able to explain the

widespread and continued decline of SSL in the larger river channels in Germany. This notion is further supported by the constant SSC level between 1970 and 1990 (Fig. 8) and the 'delayed' decline, which mainly starts around 1995 (see Fig. 8) two decades after the construction of the last dams in the German waterways.

Little is known about the evolution of soil erosion and sediment supply to river systems in Germany in the last century including the time between 1990 and 2010. Fig. 9 suggests that there was a maximum of sediment supply to the Rhine system sometime



between 1750 AD and 1970 AD. Furthermore, the magnitude of that maximum is not known. It is very likely that the construction of barrages in the river systems and reservoirs in the headwater river systems resulted in an initial decline in the middle of the last century (Fig. 9). However, the decline of the SSC after 1990 must be related to decreasing sediment supply from hillslopes (see discussion on potential sources of declining sediment supply above)

While the construction of large reservoirs and dams ceased by the 1980s, many small-scale rainwater retention basins were

built between 1990 and 2010 to reduce the flooding potential of small to medium sized creeks and rivers. The trapping efficiency of such a single small-scale feature is certainly low, but given the large number of the features, their overall cumulative effect might be significant for the reduction of suspended sediment in the large rivers. Furthermore, artificial (urban) surfaces strongly increased between 1990 and 2010 (Tab. 1) and waste water treatment plants that collect surface runoff from these surfaces became more effective during the same time. Thus, an increasing fraction sediment is trapped in

stormwater reservoirs and by waste water treatment. Therefore, we argue that the widespread and consistent trend of declining SSC in the large German river systems is mainly driven by the expansion of conservation agriculture and the increasing number of headwater retention basins in Germany.

## 5 Conclusion

Based on daily monitoring of SSC along large German river channels, we found significant declines of mean annual average

SSC at 47 of 49 monitoring stations between 1990 and 2010. On average SSC declines by -0.92 mg l$^{-1}$yr$^{-1}$. At some stations decreases during the 20 years represent up to 50% of the long-term average SSC. Significant decreases of SSC are associated with declining SSL loads.

In the context of long-term changes of suspended sediment transport, the contemporary suspended sediment loads of the Rhine at the German-Dutch border approaches the pristine level of ~1 Mt yr$^{-1}$, which was achieved by the Rhine during the mid-

Holocene when the suspended sediment load was adjusted to the Holocene climatic conditions and before the onset of increased loads due to human induced land use changes in the Rhine catchment.

The coherent and strong decline of suspended sediment concentrations and loads is difficult to explain by a mono-causal response to a change of a single driving factor. At this point we can only speculate regarding potential reasons for a decline in sediment supply to larger rivers. We argue that changes in soil erosion within the catchments and/or the sediment connectivity

in upstream headwaters, e.g. due to the construction of small rainwater retention basins, are the major reason for declining SSC in the studied river channels. In contrast to the unclear situation in the catchments, there were little changes in the construction of dams and reservoirs along the large river channels during the observation period 1990-2010. Therefore, changes along the rivers seemed to be less important for the change in SSC. However, it remains questionable if increasing rainfall extremes during the last years and which is predicted to be continued in the next decades will shift the trends of SSC again and

result in increasing SSC levels in large German river channels after decades of SSC decline.



**Acknowledgements**

The data used in this paper provided by the suspended sediment monitoring network of the German waterways that was established in the 1960ties by the Federal Waterways and Shipping Administration (Wasserstraßen- und Schifffahrtsverwaltung des Bundes, WSV). We acknowledge the WSV for maintaining the monitoring network and for suspended sediment sampling. Furthermore, we thank XY and XY anonymous reviewer for their helpful comments and suggestions that greatly improved the quality of this paper.

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



**680 Figures**

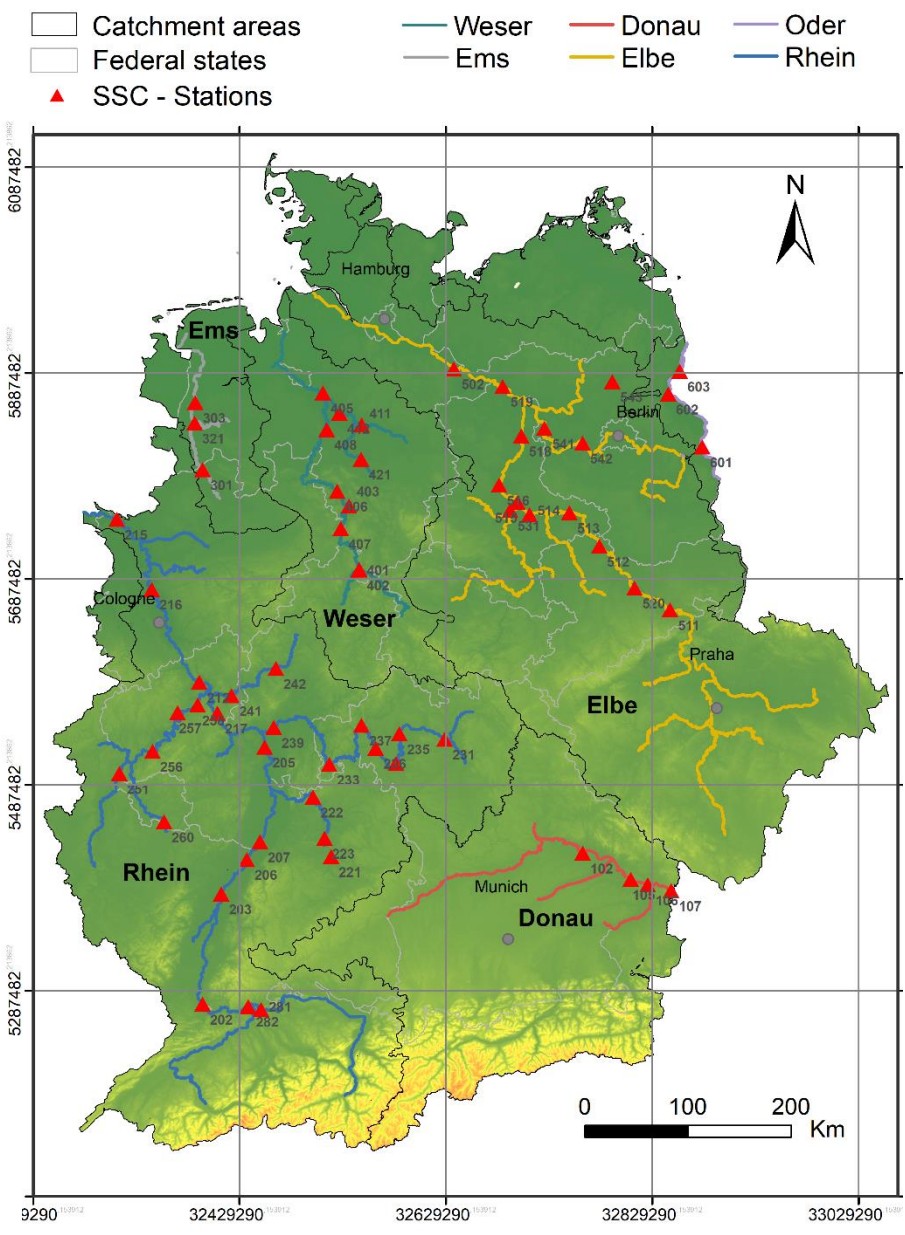

**Figure 1: Spatial distribution of sampling stations and major river systems and catchment boundaries in Germany.**





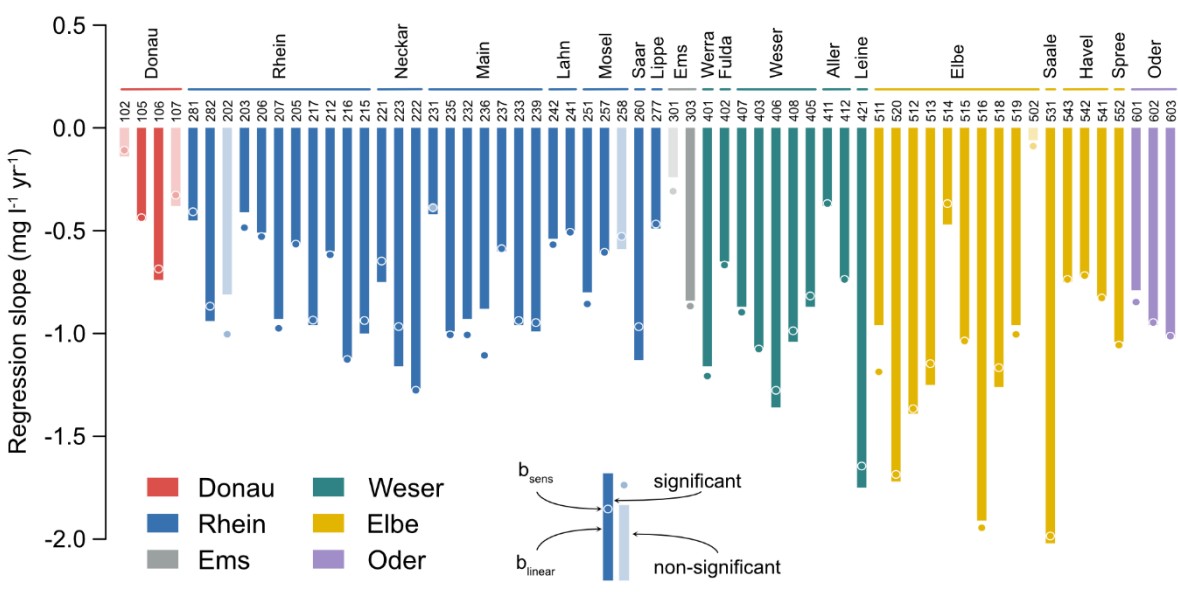

**Figure 2: Overview of SSC-trends for all 62 stations showing the trends from least-square linear regression and Sen's slope**

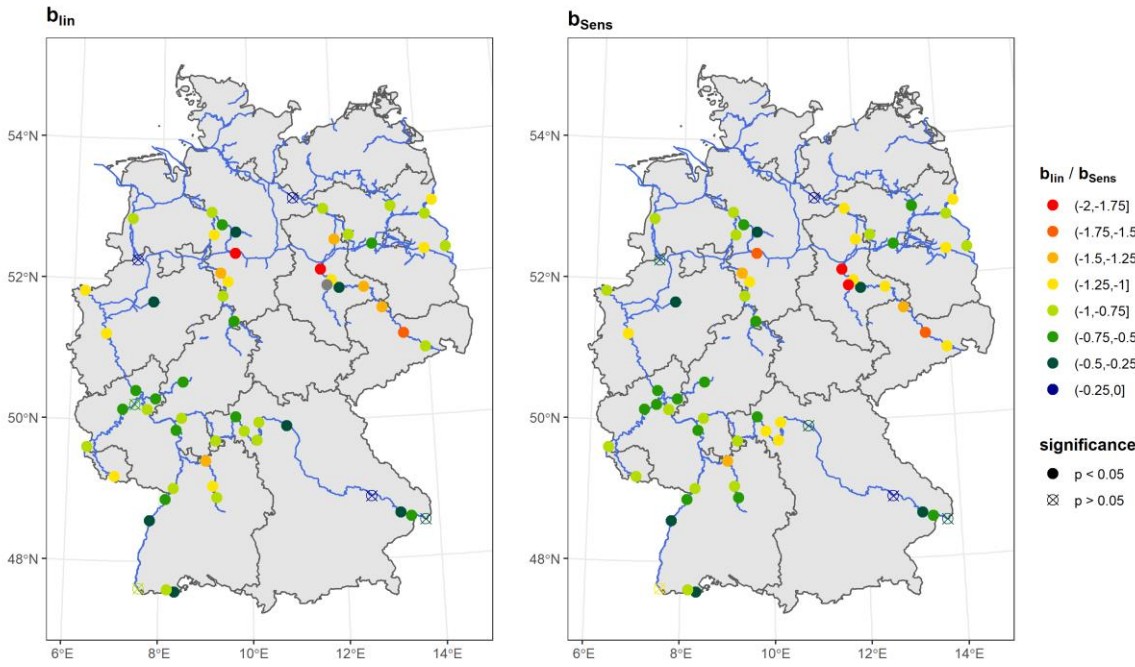

**Figure 3: Spatial distribution of trends of annual average SSC (in mg l$^{-1}$yr$^{-1}$) between 1990 and 2010 along the waterways in Germany. Left map shows trend derived from linear regression, right maps shows trend magnitude derived from Sens' slopes.**





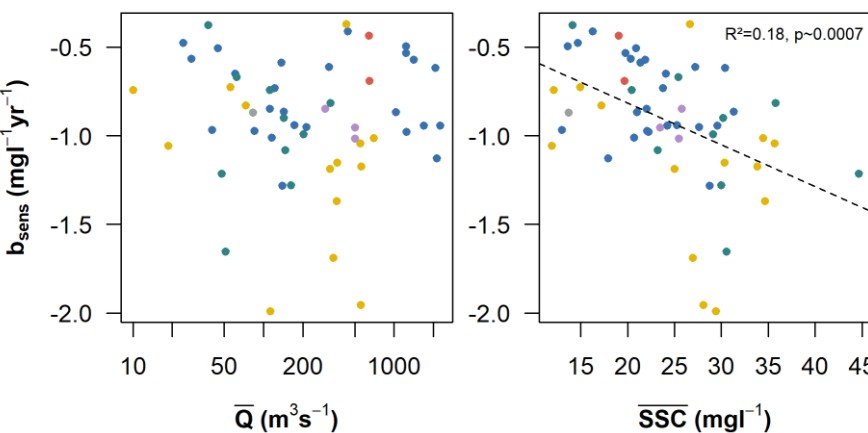

**Figure 4: Sens slope for SSC of all stations as a function of mean discharge and mean SSC.**


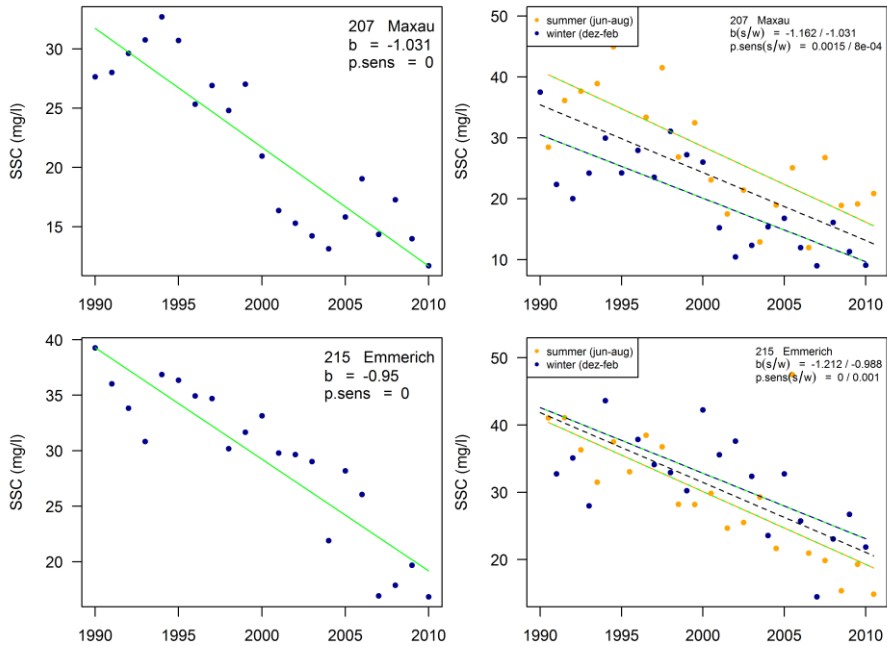

**Figure 5: Examples of SSC trends between 1990 and 2010 for monitoring station Maxau (top) and Emmerich (bottom). Left panel shows annual average SSC, right panel show trends for summer and winter months.**



Earth **Surface**
**Dynamics**
Discussions



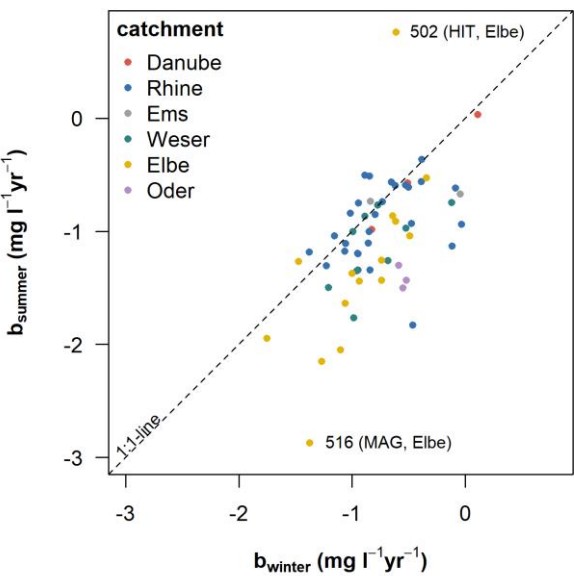


**Figure 6: Scatterplot of Sens' slope for winter and summer months.**

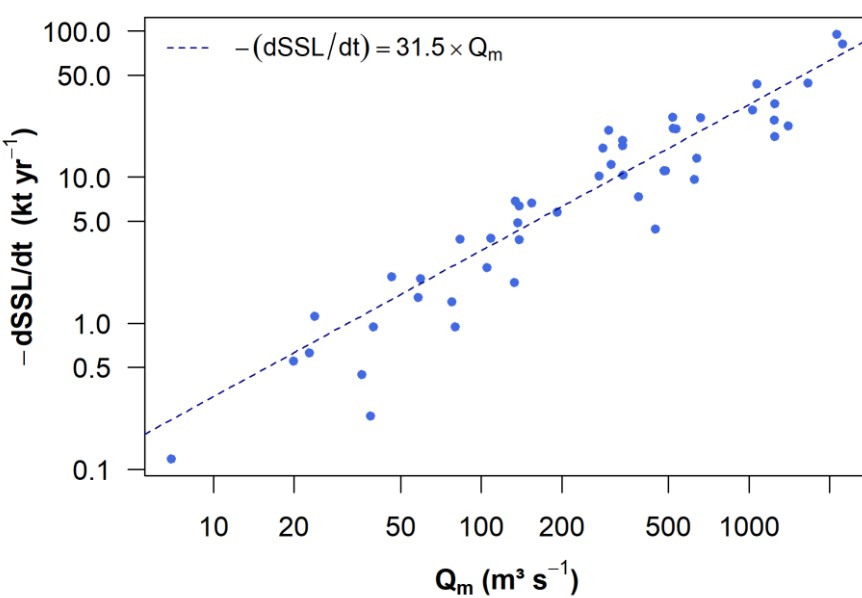

**Figure 7: SSL-decline (-dSSL/dt) as a function of discharge for all stations with a significant change of SSL between 1990 and 2010.**
**Linear regression**



Earth **Surface**
**Dynamics** Open Access
Discussions
EGU

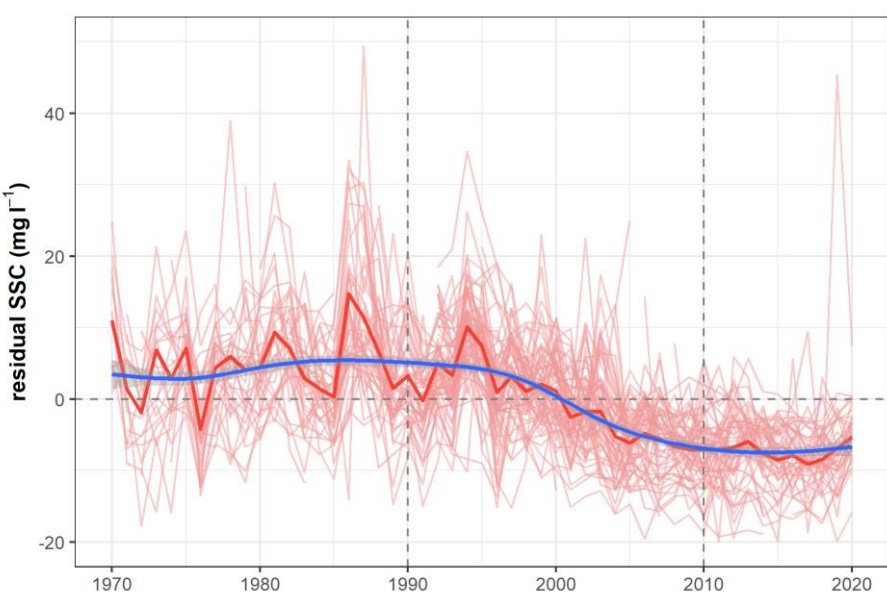

**Figure 8: Trends of residual SSC between 1970 and 2020. SSC residuals are calculated based on daily SSC data and the average**
**SSC of all available SSCs covering the full monitoring length for each station. Light red lines indicate the annual average of SSC**
**residuals for each station. The bold red line represents the average annual SSC residual of all stations. The blue line represents a**
**smooth spline of the average annual SSC residuals.**

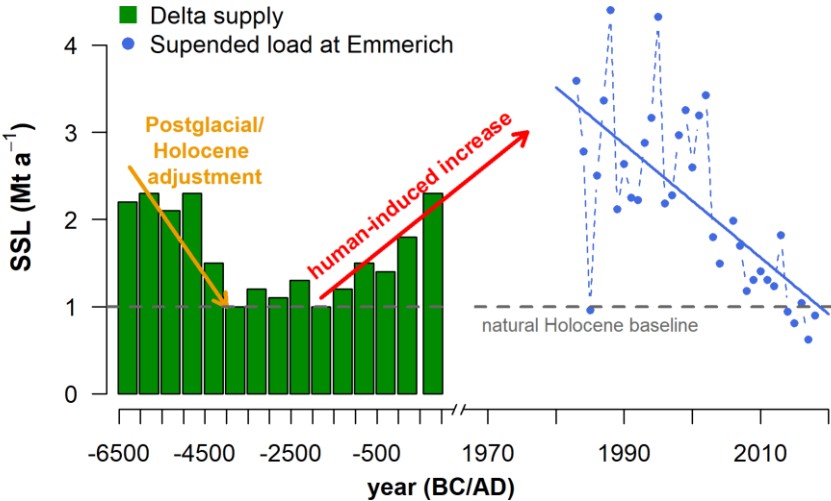

**Figure 9: Reconstructed and monitored suspended sediment loads (SSL) of the Lower Rhine at the Dutch/German border.**
**Reconstructed loads (green bars) are derived from a long-term sediment budgets analysis of the Rhine-Meuse delta in the**
**Netherlands (Erkens, 2009). Blue points represent the suspended annual loads at the monitoring station in Emmerich (representing**
**the most downstream station of the German suspended monitoring network).**



Earth **Surface**
**Dynamics**
Discussions



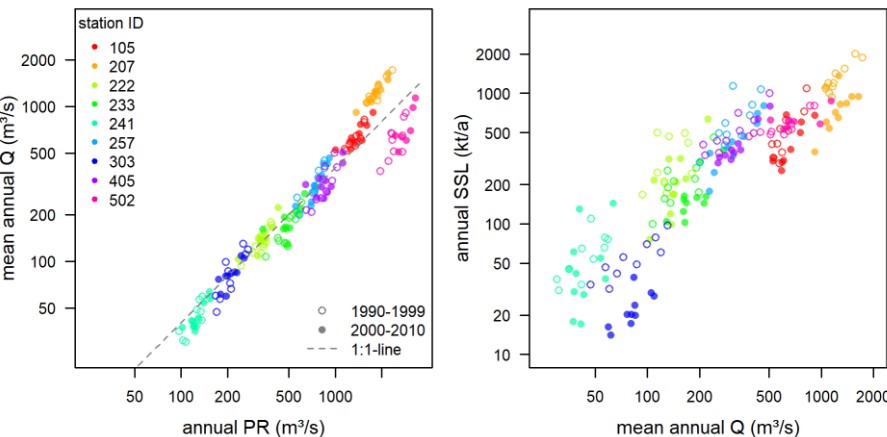


**Figure 10: Mean annual discharge as a function of annual precipitation (left graph) and annual suspended sediment load (SSL) as a function of mean annual discharge (right graph) for nine selected monitoring stations between 1990 and 2010. For station IDs see Tab. S1 and Fig. S1.**

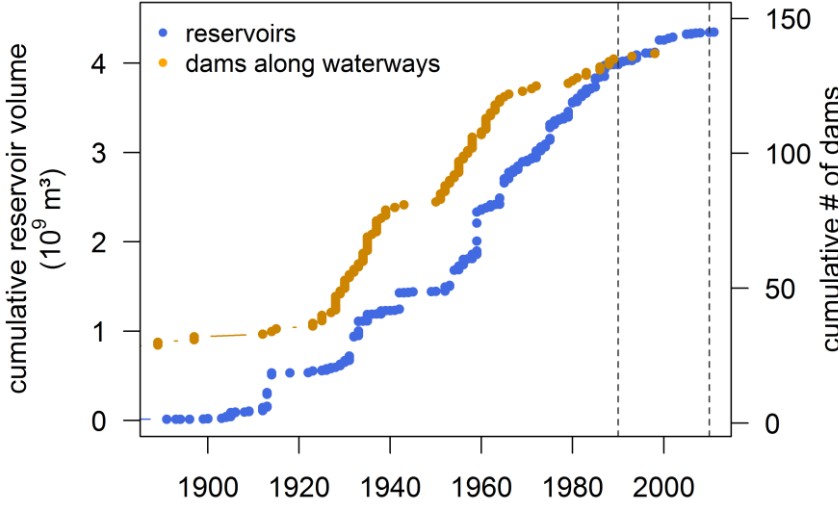


**Figure 11: Cumulative volume of reservoirs in Germany and number of dams (barrages) along the German waterways since the start of construction in the 1880ties. The cumulative volume of reservoirs includes only large reservoirs with volumes > 0.04 Mio. m³ and is taken from Wikipedia ("Liste der Talsperren in Deutschland", accessed at January 2021)**



**Tables**

**Table 1: Fraction of land cover classes derived from the CORINE land cover data for major river catchments in Central Europe (© European Union, Copernicus Land Monitoring Service 2022, European Environment Agency (EEA)).**

| year | Danube | Rhine | Neckar | Main | Lahn | Moselle | Ems | Weser | Elbe |
|------|--------|-------|--------|------|------|---------|-----|-------|------|
| arable land | | | | | | | | | |
| 1990 | 27,8 | | 30,7 | 36,8 | 25,2 | 24,5 | 65,5 | 43,3 | 47,6 |
| 2000 | 27,5 | 21,9 | 30,3 | 36,4 | 24,3 | 23,7 | 64,3 | 42,4 | 44,2 |
| 2006 | 27,3 | 21,9 | 30,1 | 36,0 | 24,1 | 24,1 | 63,4 | 41,9 | 43,5 |
| 2012 | 31,9 | 20,7 | 34,6 | 37,9 | 21,9 | 24,8 | 63,2 | 39,7 | 40,5 |
| 2018 | 31,9 | 20,7 | 34,5 | 38,1 | 21,9 | 24,8 | 63,2 | 39,6 | 40,4 |
| Forest | | | | | | | | | |
| 1990 | 34,0 | | 35,3 | 37,9 | 43,0 | 36,5 | 13,0 | 33,7 | 29,9 |
| 2000 | 33,3 | 35,6 | 34,6 | 37,9 | 43,1 | 36,5 | 13,0 | 33,7 | 30,6 |
| 2006 | 33,1 | 35,3 | 35,0 | 38,1 | 43,1 | 36,4 | 13,0 | 33,8 | 30,8 |
| 2012 | 34,4 | 35,0 | 37,0 | 39,6 | 44,3 | 37,5 | 14,0 | 34,5 | 31,6 |
| 2018 | 34,4 | 35,1 | 36,9 | 39,6 | 44,2 | 37,6 | 14,0 | 34,5 | 31,7 |
| Pasture | | | | | | | | | |
| 1990 | 14,4 | | 8,7 | 6,3 | 16,5 | 18,4 | 3,0 | 8,4 | 5,5 |
| 2000 | 14,5 | 11,4 | 8,6 | 6,3 | 16,7 | 19,2 | 3,2 | 8,8 | 8,0 |
| 2006 | 14,3 | 11,3 | 8,6 | 6,3 | 16,3 | 19,1 | 3,2 | 8,7 | 8,5 |
| 2012 | 20,2 | 11,9 | 15,1 | 14,2 | 23,4 | 21,1 | 10,5 | 14,8 | 12,6 |
| 2018 | 20,1 | 11,9 | 15,1 | 14,0 | 23,3 | 21,0 | 10,4 | 14,8 | 12,5 |
| artifical (urban & industial) surface | | | | | | | | | |
| 1990 | 5,4 | | 9,5 | 5,2 | 7,2 | 6,9 | 6,6 | 7,0 | 7,1 |
| 2000 | 5,8 | 7,5 | 10,3 | 5,5 | 7,5 | 7,2 | 7,5 | 7,4 | 7,3 |
| 2006 | 6,1 | 7,8 | 10,7 | 6,0 | 8,0 | 7,7 | 8,1 | 7,8 | 7,4 |
| 2012 | 6,9 | 8,2 | 11,9 | 6,9 | 9,0 | 8,6 | 10,4 | 8,8 | 7,8 |
| 2018 | 7,0 | 8,3 | 11,9 | 6,9 | 9,0 | 8,7 | 10,4 | 8,8 | 7,9 |
| remaining area | | | | | | | | | |
| 1990 | 18,4 | | 15,8 | 13,9 | 8,1 | 13,6 | 11,8 | 7,7 | 10,0 |
| 2000 | 18,9 | 23,6 | 16,1 | 13,9 | 8,4 | 13,4 | 11,9 | 7,7 | 9,9 |
| 2006 | 19,1 | 23,7 | 15,6 | 13,7 | 8,6 | 12,6 | 12,3 | 7,9 | 9,8 |
| 2012 | 6,6 | 24,1 | 1,5 | 1,4 | 1,5 | 8,0 | 1,9 | 2,2 | 7,6 |
| 2018 | 6,7 | 24,1 | 1,5 | 1,4 | 1,5 | 7,9 | 1,9 | 2,2 | 7,5 |