# Peer review of "Back to pristine levels: a meta-analysis of suspended sediment transport in large German river channels."

_Earth Surface Dynamics, 2022_

## Author Comment (AC1)

**Revision of the manuscript "Pristine levels of suspended sediment in large German river channels during the Anthropocene?"**

(reply by the authors in italic letters)

**Comment to AE**

*We are thankful for the two very positive reviews on the manuscript. Both reviewers provided constructive comments that greatly improved the quality of the revised paper. In addition to the reviewers comments we were able to improve our analysis regarding the development of soil erosion in Germany between 1990 and 2010. In the original manuscript we combined information from available literature regarding the development of the R-factor of the USLE and changes of land cover derived from the CORINE dataset to argue about likely changes of soil erosion in Germany. Due to discussion we Karl Auerswald, one of the leading scientists on soil erosion in Germany, we were able to calculate changes of soil erosion in Germany between 1990 and 2010. Therefore, we used data from statistical year books from Germany to derive the land cover of erosion-intensive crop types (mainly maize, potatoes and sugar beets) that are characterized by specific C-factors. A similar analysis was not possible based on the CORINE data. We were further able to calculate specific trajectories of the C-factor in East and West Germany due to there variable developments of the land use management in both parts after the reunification of Germany in 1990. This new analysis is also in line of the request from RC2 asking for a better comparability of different developments of SSC in various stations. Furthermore, it supports our statement that erosive hillslopes must be strongly decoupled from sediment transport in large river system, highlighting towards the crucial role of the management of medium sized river systems, which must act as major sediment sink.*

*To avoid the potential misunderstanding that Germany waterways are in a pristine state due to their reduced suspended sediment transport, we change furthermore the title of the submitted publications to "Pristine levels of suspended sediment in large German river channels during the Anthropocene?" We basically argue that suspended sediment load of the Rhine River is similar to the load during the Mid-Holocene, which resembles a pristine state. However, this similarity is certainly not due to a similar functioning of the river system during modern und Mid-Holocene times.*

*We hope that the AE and the Reviewers agree with the supposed changes and are convinced that these changes increased the value of the manuscript.*

*Kind regards*
*On behalf of the author team*

*Thomas Hoffmann*

**RC1: Review of "Back to pristine levels: a meta-analysis of suspended sediment transport in large German river channels"**

This paper presents new analysis on suspended concentration changes in many German rivers during the 1990-2010 period. The authors show that there is a significant decrease in surface suspended concentrations by an average of -0.92 mg/L/yr related to a decrease of sediment supply. Interestingly, the suspended sediment load is (in 2010) very close to the reconstructed Early Holocene sediment load before human impact on the landscape. Although it is difficult to single out the cause of this observed sediment decline, the authors argue that the most likely explanation is a change of agricultural practices, i.e. the expansion of conservation agriculture and increasing building of retention basins. Overall, this is a very interesting and well-written paper, with robust statistical analysis and arguments. This manuscript is worth being published in ESurf providing some minor revisions (see below).

*Thanks for this very positive review and comments on our manuscript. All specific comments will be considered and are helpful to improve the manuscript. Some comments do not apply anymore due to the changes made in the revision (see reply to reviewer RC2).*

**Specific comments:**

- Lines 34-47: this paragraph is a bit too long. I suggest reducing it by 50%
  → *We strongly reduced the length of this paragraph.*
- Line 60: "In 2009/10 about 38% of arable land were under soil conservation" should be "In 2009/10 about 38% of arable land was under soil conservation"
  → *this paragraph was removed*
- Line 65 "periode" should be "period"
  → *this paragraph was removed*
- Lines 100-101: what is the pore size the coffee filters that were used?
  → *That is a very valid question. We added some information in the manuscript 'However, these filters do not have a well-defined pore diameter and a significant fraction of clay is lost (compare Hoffmann et al. 2020).', but refer to Hoffmann et al. (2020) for further information.*
- Line 109: is there a rationale for choosing 150 sample/year as the cutoff number of samples for considering annual SSC representative? Or is it a value chosen arbitrarily? (Which is fine, but maybe specify it in the main text).
  → *We added additional information motivating the choice of 150 samples: '…(representing approx. 50% of the samples taken during a year at a station)…'. The 50%, however, is arbitrarily chosen.*
- Lines 130-132 and 170-171: what are the reason(s) for not using rating curves to correct for missing measurements? A sensitivity test (e.g. in the supplementary materials) for a few rivers comparing mean SSC with and without correction for missing measurements using rating curve approach and discharge may be useful here to demonstrate that the data gaps do not affect the calculated mean SSC and observed trends. Also, are the 150 data points evenly distributed throughout seasons for all rivers?
  → *Thanks for these very good questions. In case of existing trends in the dataset, it is not valid to use a single SSC/Q rating curve as the SSC/Q-relationship changes with time. Therefore, filling need to be done using annual SSC/Q-rating curves, limiting the number of measurements to constrain the rating relationship and may thus introduce additional errors. As correctly suggested by reviewer 1, we added a simple sensitivity analysis for three stations in Figure A1. Trends are calculated using non-gap-filled*

*data (left column in Fig A1) and gap-filled data using annual rating based on loess regression (right column in Fig. A1). The results reveal marginal differences of the SSC trends based on the Sens-slope.*

- Line 141 "the assumption of LSR is violated" It would be clearer if you stated what this assumption was first in case people are not familiar with linear least squares regression

  → *We rephrased the sentence to clarify: '12 stations show non-normally distributed residuals of the LSR, indicated by italic $p_{lin}$-values in Tab. S2.'*

- Line 184 "Warwick (2015)" should this be "Warrick (2015)" as before? If not and it is a separate reference, it is missing from the reference list

  → *changed to Warrick (2015)*

- Line 214 "artificial areas" I see this is how CORINE names this group of classes (which contains Urban fabric, industrial units, mining and construction sites, artificial vegetated areas) but the name itself is a bit vague. Maybe at line 216 you could specify the kind of land cover class this refers to.

  → *We removed CORINE land cover data from this manuscript and used data from the German Statistical Yearbooks between 1990 and 2010. These provided a better differentiation of the crop types and allowed for rough C-factor calculation between 1990 and 2010 (see also reply to reviewer R2)*

- Lines 247-248: in figure 4, is the "average SSC" the average during the 1990-2010 period?

  → *correct, we added details in the text and in the figure caption.*

- Lines 303-305: this is important. Many large river studies have shown that SSC increases (sometimes by a factor 2 to 5) with depth. This increase SSC with depth often related to increase of suspended silt and sand content above the riverbed. Is there any existing study on the Rhine or other German rivers reporting the extent of possible suspended sediment concentration change with depth? Although probably unlikely, a change in the proportion of fine/coarse sediment supply could lead to a decrease of surface SSC (fine sediments) but increase of SSC at depth and therefore an increase of depth average SSC?

  → *The reviewer raises an important question about the vertical distribution of suspended silt and sand. The German water and shipping authority maintain a monitoring network, which considers the spatial variability of suspended sediments in river cross sections and measures the sand concentrations at various depths based on 50l water samples. The sampling interval at a location varies between 1-3 measurements a year and cannot be used for a statistically sound trend analysis, however, it allows to infer that the Csand/Ctot-ratio did not significantly changed during the period from 1990 to 2010 at three stations along the river Rhine (see Figure below). We therefore argue that there was no general change in the proportion of the fine/coarse sediment supply.*

[Figure]

- Lines 322 and 325: technical correction, replace "Fig. 4" by "Fig. 5"
  - → *Figure number all changed and are mofied accoringly*
- Line 349 "decreases sediment loads had no major impact" should be "decreases in sediment loads had no major impact" or "decreasing sediment loads had no major impact"
  - → *done*
- Line 390 "SSCs changes between 1970 and 2020 are unraveled by the compilation of the residual SSCs which were calculated using the daily SSCs and the long term average SSC for each station." This is not really clear. Are these the residuals from your Sen slope regression?
  - → *we rephrased this section to clarify that the residuals are the differences between the annual mean SSCs and the long-term average SSC of each station.*
- Figure 11: The dashed lines show the start and end of your study period. It might be clearer if you labelled them.
  - → *Fig. 11 is updated as suggested with labels for dashed lines*

**RC2: Review of Hoffmann et al "Back to pristine levels: a meta-analysis of suspended sediment transport in large German river channels"**

In this manuscript Hoffmann and colleagues analyse the suspended sediment concentration (SSC) and load of 9 large river catchments in Germany. Over a 20-year period (1990 – 2010) they find that nearly all (49/62) of the measurement stations recorded a continual decline in mean annual SSC. By the end of the measurement period the SSC seems to be closing in on a natural base level previously observed before human activity in the Rhine valley. The authors then analyse multiple possible driving processes for this trend. This analysis is thorough but they are not able to propose a single likely process driving this trend and so instead suggest a combination of smaller factors, such as soil erosion controls and small-scale flood defences may contribute to the observed trend.

I enjoyed reading this manuscript and found it very thought provoking. I thought the analysis on the whole is thorough and well done, but I feel some more comparison between the basins/stations could offer some insight into the proposed controlling factors. For example, catchments with higher arable land use percentages could be more impacted by any change in soil conservation practices while catchments which have seen flooding in the recent past are more likely to be targeted for flood defences such as rainwater retention basins potentially resulting in a greater decline. Comparative analysis between the catchments may offer some more support for the closing arguments which is currently lacking. I believe these changes, and some other line by line comments I make below, would be minor in nature.

→ *Thank very much for your valuable and positive feedback. This is very much appreciated. We fully agree that a better comparison of the station data should yield more insight into the controlling factors. Therefore, we changed the discussion on light of the land use changes that took place after the reunification of Germany in 1990. Major differences in the land use change took place in the Western und Eastern part of Germany that belong to the Federal Republic and the German Democratic Republic, respectively. We used these differences to calculate the expected developments of the C factor of the universal soil loss equation (USLE) in West and East Germany. We based this analysis on the Statistical Yearbook of Germany in which land use type and management is summarized for each state. Therefore, we removed the analysis of the CORINE land cover data, which are spatially distributed, but which are less precise, especially with respect to the different crop types, that we used to reconstruct the changes of the crop-factor (C-factor) of the USLE. Furthermore, we were able to calculate the development of soil erosion in Germany between 1990 and 2010 based on the combined changes of the rainfall erosivity and C-factor. Karl Auerswald strongly supported us with this analysis. Therefore, we added him as a new co-author to the revised manuscript.*

*Despite major differences of the C-factor development between West and East Germany between 1990 and 2010, the declining trends of stations located in West and East Germany do not differ, supporting our hypothesis, that the SSC decline in large river channels in Germany is dominantly controlled by the decreasing sediment connectivity between eroding hillslopes and large river channels.*

**Line by line comments:**

Line 44: Waterways may need a definition
→ *done*

Line 63: Rainfall erosivity needs a definition here
→ *this paragraph was removed, but rainfall erosivity was explained in detail later in the MS*

Line 63: Periode should be period
→ *this paragraph was removed*

Line 65: The original area could be included to add some further contextualisation about the impact of this change.
→ *this paragraph was removed*

Line 72: It is not immediately clear what is meant by Work-daily. Perhaps a short definition is required.

→ *we changed 'work-daily' to 'daily' here. Later in the second paragraph of section 2.1 we explain that sampling is limited to work-days.*

Line 144: It is not clear why the Mann-Kendall test is being used or what it tests from this description.
→ *We added more information to clarify: 'To evaluate the limitations of the linear regression on time series data we additionally used the non-parametric Mann-Kendall test to detect monotonic trends of SSC for each station. In contrast to LSR, the rank-based Mann-Kendall test does not rely on normally distributed residuals and does not make any assumptions about the type of trend (i.e. linear or non-linear) as long as values are changing monotonically.'*

Line 145: I also found the description of the Sen's slope unclear. What is the magnitude of the trend?
→ *We rephrased the sentence to explain the magnitude of the trend: 'In case of significant trends at a 5% level (i.e. Mann-Kendall's p<0.05), we estimate the magnitude of the annual SSC trend in mgL-1a-1using the Sen's slope'*

Line 145: Calculate should be calculated
→ *done*

Line 171: "identify changes is suspended…" should be identify changes **in** suspended…
→ *done*

Line 172: Units of these metrics would help to explain them.
→ *we added units*

Line 173: I am not sure what is meant by "reactivity of river catchments"
→ *we reworded this paragraph and avoided the term 'reactivity'.*

Line 180: When is the log-linear regression analysis used?
→ *we reworded the previous sentence to clarify why log-linear regression is used.*

Line 184: Missing word: "Surface **runoff** generating rain fall events".
→ *done*

Line 202: rainfall erosivity still needs a definition.
→ *we added details regarding the definition of rainfall erosivity (including a reference) and regarding its calculation*

Line 231: Why is there no change seen at these 6 stations? Could there be any information derived from these stations? (and again on lines 255 and 260 for the seasonal data)
→ *This is a very good question/recommendation. In addition to the discussion of the Hitzacker station that shows no change and is already discussed in section 4.1, we added a full paragraph discussing no change at five of the six stations in section 4.1.*

Line 270: Do these analysis techniques identify the same stations as not changing?
→ *we added additional information subsequent to the sentence, which is describing no change for rating coefficient a.*

Line 349: "Decreases **in** suspended loads"
→ *done*

Line 366: "into the Black **Sea**"
→ *done*

Line 378: TOC needs a definition
→ *done in Line 372*

Line 419: The reference to Figure 7 should be to figure 9.
→ *correct, changed to the new (correct) Figure number*

Line 423: Is the rate of the decline between 1990 and 2010 worth discussing? It is 10x times faster than what is potentially seen in the stratigraphy.
→ *... and approx. 100 times faster than the increase from the mid Holocene. We used this number to relate the rate of change in this context. This might not be globally representative, but we were not sure were on which evidence the '10x times faster' statement was based.*

Line 433: It is not clear what "an increase in rainfall erosivity between April and November by 2.1 % per year or 42 % from 1990 to 2010" means. Is there an increase in erosivity each year between April and November?
→ *This paragraph was completely rewritten*

Line 438: The acronym USLE is not defined
→ *USLE is now defined in section 2.4, where we define the rainfall erosivity, which is the R-factor in the USLE*

Line 455: It is not clear where this data can be seen in figure 10.
→ *This paragraph and the relevant Figure were removed*

Figure 3: The points for stations with no significant difference are not always easily visible
→ *We increased the stroke of the insignificant points. They are now much easier to see.*

---

## Author Response (AR2)

**Technical corrections to the manuscript "Pristine levels of suspended sediment in large German river channels during the Anthropocene?"**

Dear Tom and Valier,

Thanks very much for your positive feedback on our manuscript. We addressed all minor comments in the annotated PDFs from the AE and addressed the to questions raised by the AE:

L94-95: Please expand on the potential effect of loosing a portion of the clay-size fraction in the context of the observed trends. Is this expected to reduce or increase the magnitude of the trends given the suspected underlying mechanisms?

➔ We added a statement that we are convinced that the loss of e proportion of the clay-size fraction did not affect the calculated trends, as the production of the filters did not change during the monitoring period.

➔

L361-363: For this explanation to hold, the contribution of organic matter to SSC must be quantitatively significant. Is this supported by measurements of the organic carbon concentration in suspended sediments at this station?

➔ We added a statement including the average share of 28% of phytoplankton to the total suspended sediments.

Kind regards

Thomas (Hoffmann)